# The Effect of Pre-Operative Verbal Confirmation for Interventional Radiology Physicians on Their Use of Personal Dosimeters and Personal Protective Equipment

**DOI:** 10.3390/ijerph192416825

**Published:** 2022-12-15

**Authors:** Satoru Matsuzaki, Takashi Moritake, Lue Sun, Koichi Morota, Keisuke Nagamoto, Koichi Nakagami, Tomoko Kuriyama, Go Hitomi, Shigeyuki Kajiki, Naoki Kunugita

**Affiliations:** 1Department of Radiology, Shinkomonji Hospital, 2-5 Dairishinmachi, Moji-ku, Kitakyushu 800-0057, Japan; 2Department of Radiation Regulatory Science Research, National Institute of Radiological Sciences, National Institutes for Quantum Science and Technology, 4-9-1 Anagawa, Inage-ku, Chiba 263-8555, Japan; 3Health and Medical Research Institute, Department of Life Science and Biotechnology, National Institute of Advanced Industrial Science and Technology (AIST), Central 6, 1-1-1 Higashi, Tsukuba 305-8566, Japan; 4Department of Radiology, Hospital of the University of Occupational and Environmental Health, Japan, 1-1 Iseigaoka, Yahatanishi-ku, Kitakyushu 807-8556, Japan; 5Department of Occupational and Community Health Nursing, School of Health Sciences, University of Occupational and Environmental Health, Japan, 1-1 Iseigaoka, Yahatanishi-ku, Kitakyushu 807-8555, Japan; 6Department of Radiological Technology, Kawasaki Medical School Hospital, 577 Matsushima, Kurashiki 701-0192, Japan; 7Department of Occupational Health Practice and Management, Institute of Industrial Ecological Science, University of Occupational and Environmental Health, Japan, 1-1 Iseigaoka, Yahatanishi-ku, Kitakyushu 807-8555, Japan

**Keywords:** interventional radiology, personal passive dosimeter, personal protective equipment, direct intervention, pre-operative briefing

## Abstract

Interventional radiology (IR) physicians must be equipped with personal passive dosimeters and personal protective equipment (PPE); however, they are inconsistently used. Therefore, we aimed to explore practical measures to increase PPE usage and ascertain whether these measures could lead to an actual decrease in exposure doses to IR physicians. Dosimeters and PPE were visually inspected. Then, a pre-operative briefing was conducted as a direct intervention, and the use of dosimeters and PPE was verbally confirmed. Finally, the intervention effect was verified by measuring the use rates and individual exposure doses. Because of the intervention, the use rate markedly improved and was almost 100%. However, both the effective dose rate (effective dose/fluoroscopy time) and the lens equivalent dose rate (lens equivalent dose/fluoroscopy time) showed that the intervention led to a statistically significant increase in exposure (effective dose rate: *p* = 0.033; lens equivalent dose rate: *p* = 0.003). In conclusion, the proper use of dosimeters and PPE raised the radiation exposure values for IR physicians immediately after the intervention, which was hypothesized to be due to the inclusion of exposure overlooked to date and the changes in the dosimeter management method from a single- to a double-dosimeter approach.

## 1. Introduction

Interventional radiology (IR) has undergone technological innovations in many specialized fields. It has become an essential medical procedure, not only for saving patients’ lives, but also for ensuring post-treatment quality of life. However, occupational exposure of healthcare workers involved in IR has become an issue [1]. The objectives of radiological protection are to manage and control exposures to ionizing radiation so that tissue reactions are prevented. The risks of stochastic effects should be reduced to as low as reasonably achievable, with societal and economic factors considered [1,2,3]. To achieve these objectives, IR physicians must wear lead glasses to avoid cataracts, which are caused by a tissue reaction to a threshold dose. Moreover, although the risks of cancer, a stochastic effect with a linear non-threshold dose–response relationship between the dose and risk, cannot be reduced to zero, physicians are required to wear a lead apron and a thyroid protection collar to minimize the risks as much as possible.

For healthcare professionals to conduct radiological procedures while reducing their ionizing-radiation exposure, it is crucial to introduce the following forms of management for occupational health: general management, work environment management, work management, health management, and occupational health training [4,5,6] (Table 1). However, it is difficult to establish whether these requirements are carefully practiced in the healthcare industry in Japan [7]. Radiological procedures are usually conducted without special radiation protection and without education or training depending on the exposure levels [3,8,9].

When conducting radiological procedures, it is necessary to wear a personal passive dosimeter for exposure dose evaluation and personal protective equipment (PPE) for physical protection; PPE for ionizing radiation includes a lead apron, lead glasses, and a thyroid protection collar. According to a previous questionnaire survey based on self-response, the use rate was 17–100% for personal passive dosimeters [7,10,11,12,13,14,15], 88–100% for lead aprons [11,13,14,15,16,17,18], 0–83% for lead glasses [10,11,12,13,14,15,16,17,18], and 40–100% for thyroid protection collars [10,14,15,16,17,18]. Various use-rate values that range from high to low have been reported for equipment other than the lead apron (Table 2), and there is a possibility that proper personal dosimetry and radiation protection may not be employed. Improving this situation requires education and training that is specialized to the exposure circumstances of physicians. However, studies that analyzed the effects of radiation protection education to date [14,19,20] have indicated that physicians’ use rate of personal passive dosimeters has not reached 100% even after implementing radiation protection education [14]. Although there have been studies indicating that personal exposure for physicians decreased after conducting radiation protection education [19,20], if the personal passive dosimeter use rate is not 100% to begin with, then any shifts in the personal exposure dose cannot be accurately evaluated. Thus, it is expected that the usual radiation protection education, which is provided to physicians initially and while they continue radiation work, will be insufficient, and some additional direct intervention is essential for improving the environment of physicians who work in busy clinical settings.

In 2009, the World Health Organization (WHO) recommended the “WHO Guidelines for Safe Surgery” [21] and published the Surgical Safety Checklist for improving the safety of surgical patients. This checklist was customized for each clinical area and helped avoid adverse surgery events and reduce human error [22,23,24,25]. Pre-operative briefings and “time-outs” have been introduced as standard procedures for surgical care [26]. Pre-operative briefings are conducted to facilitate team communication, facilitate important patient communication regarding surgery, and share learning and responsibilities for surgery [21]. Furthermore, “time-outs” are conducted to avoid simple human error by having the entire team stop work for a short period immediately before surgery to check patient information and the surgery site [21]. These concepts can be applied to surgical procedures and helps to ensure the safety of patients in IR. Given that it is highly effective in reducing patient exposure doses [27,28,29,30,31,32], there are an increasing number of facilities that are introducing briefings and time-outs. However, there are still no studies of the effects of these briefings and time-outs on the status of radiation protection in the physicians who conduct IR themselves.

It is necessary to explore practical measures to increase the use of personal passive dosimeters and PPE to prevent hazardous radiation risks to IR physicians. Whether the proper use of dosimeters and PPE could lead to the actual decrease in exposure dose to physicians must be proved. Therefore, in the present study, an inspector, i.e., a radiological technologist overseeing the process, conducted an accurate survey via visual inspection of the use status of dosimeters and PPE at a major Japanese hospital. Next, briefings were conducted as direct interventions to confirm the operator’s proper use of the dosimeter and PPE. Finally, the effect of this direct intervention was evaluated.

## 2. Materials and Methods

### 2.1. Research Design

This study first looked at the actual circumstances of the use status of the IR physicians’ dosimeter and PPE via visual inspection during the pre-intervention period. Next, as a direct intervention, a pre-operative briefing was conducted before starting radiological procedures during the intervention period. The correct use of dosimeters and PPE was inspected and verbally confirmed in the briefing. If operators had any deficiencies, they were encouraged to wear the proper equipment. Finally, the effect of the intervention was mainly analyzed from two perspectives: the use status of the personal passive dosimeter and PPE, and the personal exposure dose and dose rate to the operator. 

The survey was conducted at an acute hospital with 214 beds (300–340 emergency cases/month) in Japan. Personnel changes are often made from April according to the fiscal year in Japan. Therefore, this survey was planned to start in April 2017 and end in February 2018 to prevent the physicians from leaving within the survey period. 

As shown in Table 3, April–August 2017 (five months) was set as the pre-intervention period, and October 2017–February 2018 (five months) was set as the intervention period. We surveyed all 964 (pre-intervention: 549 cases; intervention period: 415 cases) procedures (inspections and treatment using X-ray fluoroscopy) that the 14 physicians conducted during the survey periods. Of these, 340 cases in the pre-intervention period involved visual inspection of the use status of personal passive dosimeters and PPE by the inspector, a radiological technologist overseeing the process. In the intervention period, 321 cases involved visual inspection and verbal confirmation about using personal passive dosimeters and PPE by the inspector. The personal exposure doses to the physicians (i.e., effective dose and lens equivalent dose) and the exposure-related indicators (number of procedures and fluoroscopy time) were investigated using the medical records.

### 2.2. Survey of Personal Passive Dosimeters and PPE Use Rates

A survey regarding the use status of the personal passive dosimeters (Luminess Badge, NAGASE-LANDAUER, LTD., Tsukuba, Japan), frontal or coat-type lead aprons (0.25 or 0.35 mm Pb equivalent), lead glasses (0.07 or 0.35 mm Pb equivalent), and thyroid protection collars (0.28 or 0.35 mm-Pb equivalent) (Appendix A) was conducted by the inspector via visual inspection for each case. Generally, the physician chooses the PPE of their choice. Moreover, the choice often varies for each case. Regarding the personal dosimeters, the physician may forget to wear them or wear them inaccurately. Therefore, because it is difficult to grasp the true personal dosimeter and PPE wearing status by the self-administered questionnaire survey method, the direct visual inspection method was adopted in this study. The results were recorded using a radiation work checklist (Appendix A). The use rate was defined as the number of times the equipment was used divided by the number of visual inspections.

### 2.3. Survey of Personal Exposure Dose

Cases with a double-dosimeter approach used a main dosimeter under the lead apron and an additional dosimeter over the lead apron. Cases with a single-dosimeter approach used the main dosimeter under the lead apron (Appendix A). The personal passive dosimeter value was calibrated with a 1 cm personal dose equivalent.

The effective dose (E) was calculated using the following Equations (1) and (2):Double-dosimeter approach: E = 0.89 × D_main_ + 0.11 × D_additional_(1)
Single-dosimeter approach: E = D_main_(2)
where, D_main_ is the main dosimeter (under the lead apron) value, and D_additional_ is the additional dosimeter (over the lead apron) value.

The lens equivalent dose (H_eye_) was calculated using the following Equations (3) and (4):Double-dosimeter approach: H_eye_ = D_additional_(3)
Single-dosimeter approach: H_eye_ = D_main_(4)

### 2.4. Pre-Operative Briefing

A pre-operative briefing was conducted before starting procedures as a direct intervention in the intervention period. The inspector visually inspected the operators’ correct use of the dosimeter and PPE and filled the results in a radiation work checklist (Appendix A). Furthermore, the inspector verbally confirmed the physicians’ use of dosimeters and PPE, and if they were not using the protective equipment, the inspector encouraged them to wear it.

### 2.5. Statistical Analyses

BellCurve for Excel (version 3.22, Social Survey Research Information Co., Tokyo, Japan) was used for statistical analysis. The two groups were compared using two-sided Wilcoxson’s signed-rank test or two-sided Mann-Whitney’s U test. Differences with *p* < 0.05 were considered significant. 

## 3. Results

### 3.1. Use Rate of Personal Passive Dosimeters and PPE in the Pre-Intervention Period

All 14 physicians (13 men, 1 women) conducting IR at this hospital were surveyed; the average physician age was 38.9 ± 8.0 years (mean ± standard deviation: S.D.), and the average number of experience years was 12.1 ± 7.5 years (mean ± S.D.). 

The physicians’ use rate of the personal passive dosimeter and PPE before the intervention period is presented in Table 4. The use rate of the main dosimeter was low at 47% of the total, with significant individual differences (Appendix A). 

All physicians wore a lead apron (100%), but the use rates for lead glasses and thyroid protection collars were low at 37% and 52%, respectively, with significant individual differences (Appendix A).

### 3.2. Intervention Effect of Pre-Operative Briefing

Table 4 also indicates the physicians’ use rate of personal passive dosimeters and PPE during the intervention period. The mean overall use rate was 97% and 65% for the main and additional dosimeters, respectively, which was a statistically significant increase when compared to that of the pre-intervention period (main dosimeter: *p* = 0.002, additional dosimeter: *p* = 0.008). Furthermore, four physicians (g, h, i, and j) switched from a single-dosimeter approach to a double-dosimeter approach (Appendix A). The use rate of the lead apron was high at 100% even in the pre-intervention period, and the use rate did not show much change during the intervention period. However, the overall use rate of lead glasses and thyroid protection collars showed a statistically significant increase (lead glasses: *p* = 0.003, thyroid protection collar: *p* = 0.008).

Table 5 shows the personal exposure doses for physicians. The number of medical treatments in the pre-intervention period and the intervention period differed for each physician. Hence, we calculated the effective dose rate (µSv/min) and lens equivalent dose rate (µSv/min) by dividing the effective dose (mSv) and lens equivalent dose (mSv) of each physician by the total fluoroscopy time (min) for comparison. Results showed a statistically significant increase in the intervention period for both values compared to those in the pre-intervention period (effective dose rate: *p* = 0.033, lens equivalent dose rate: *p* = 0.003). Significantly, the lens equivalent dose rates for physicians h, i, and j in the intervention period were much higher than the dose rates for the other physicians (Appendix A). They were all orthopedic surgeons who routinely performed procedures using the over-table tube fluorographic imaging unit, so the patient’s body, which is the source of scattered radiation, was close to the eyes of the physician. Moreover, there was no shield between the patient’s body and the physician’s eyes. For these reasons, the lens equivalent dose rates for physicians h, i, and j during the intervention period seem to have become significantly higher than the dose rates for other physicians.

## 4. Discussion

Exposure protection is essential for physicians who conduct radiological procedures, and the International Commission on Radiological Protection (ICRP) has called for the provision of appropriate and sufficient information and training on exposure protection [3,4,8,9,33]. Nevertheless, according to self-response questionnaire surveys regarding the use rate of personal passive dosimeters and PPE by physicians to date, the use rate has been reported to be 17–100% for personal passive dosimeters [7,10,11,12,13,14,15], 88–100% for lead aprons [11,13,14,15,16,17,18], 0–83% for lead glasses [10,11,12,13,14,15,16,17,18], and 40–100% for thyroid protection collars [10,14,15,16,17,18] (Table 2). It can be observed that physicians and others are not always accustomed to using personal passive dosimeters and PPE that protect their bodies and the creation of a mechanism that encourages physicians and others to practice consistent exposure protection is urgently needed.

The “WHO Guidelines for Safe Surgery 2009,” a safety measure for surgical procedures, has been introduced in IR recently [27,28,29,30,31,32]. Aizer et al. reported that implementing radiation protection using time-outs before electrophysiology procedures for arrhythmia resulted in a 21% reduction of the air kerma-area product (P_KA_) [29]. Choi et al. reported that conducting radiation safety briefings and time-outs before X-ray irradiation for children’s central venous catheter placement using fluoroscopy reduced the P_KA_ by 79% [30]. Barakat et al. reported that conducting a pre-operative time-out protocol before endoscopic retrograde cholangiopancreatography reduced the P_KA_ by 35–48% [31]. Therefore, pre-operative briefings and time-outs have proven highly effective for patient exposure dose reduction when considering patient radiation safety in IR. However, there have not been any studies that objectively evaluated the effect that pre-operative briefings and time-outs had on radiation safety for physicians. The novelties of the present study were the survey of the use rate of personal passive dosimeters and PPE and that this was objectively evaluated by an inspector’s visual inspection rather than a self-response questionnaire. Furthermore, the use and personal exposure dose rates helped to objectively evaluate the intervention effect.

In the hospitals where the survey of the present study was conducted, the use rate of the main dosimeter, additional dosimeter, lead apron, lead glasses, and thyroid protection collar before starting the pre-operative briefing intervention had a median value of 57%, 0%, 100%, 15%, and 69%, respectively, with overall low values except for that of the lead apron. These tendencies were similar to those obtained in previous reports on physicians’ use rate of personal passive dosimeters and PPE (Table 2). Using a personal passive dosimeter and protecting the lens of the eye and thyroid, which are sensitive to radiation, are crucial for physicians who conduct IR [1,34,35]. Hence, there is a requirement to analyze why the use rate of this equipment is low and to implement appropriate measures to increase their use.

A common reason given by physicians for why they forget to wear additional dosimeters is that “they kept the additional dosimeter in their desk drawer, which they do not need in the outpatient department or ward.” However, in the present hospital, the main dosimeter and additional dosimeter were both stored side by side in the rack of the surgical room according to the ICRP recommendations [3] from before the start of the interventional experiment, and the physicians who were taking a double-dosimeter approach (physicians a–f in Appendix A) never forgot to wear an additional dosimeter. A unique initiative of the surveyed hospital was to not store the personal passive dosimeter on the rack for physicians conducting radiological procedures at multiple locations in the hospital. Instead, preparing a strap that connects both the main and additional dosimeters, as shown in Figure 1, and having the physicians carry both with them at all times prevented physicians from forgetting to wear the additional dosimeter.

During the survey of the target hospital before the start of the intervention, the use of personal passive dosimeters and PPE was at the discretion of individual physicians, and the use rate of equipment other than the lead apron was low overall. It must be acknowledged that only conducting radiation protection education once a year in a conventional hospital cannot improve the awareness of radiation protection among physicians and lead to higher usage of personal passive dosimeters and PPE. However, we succeeded in dramatically increasing the use rate of personal passive dosimeters and PPE among physicians by verbally confirming whether they were using this equipment and conducting pre-operative briefings (Table 4). 

Additionally, physicians g, h, i, and j used a single-dosimeter approach for their personal passive dosimeters. However, after starting pre-operative briefings as an intervention, they noticed that the main dosimeter value was unexpectedly high considering the relatively short fluoroscopy time (Appendix A). Then, they switched to a double-dosimeter approach during the intervention period. Therefore, it can be said that achieving a more desirable personal dose management method is also a secondary effect of this intervention. 

It has been reported that physicians paid attention to exposure when conducting pre-operative briefings and time-outs during IR, which significantly decreased P_KA_ as a result, or in other words, significantly reduced the exposure dose to the patient and operator simultaneously [29,30,31]. However, in contrast to these studies, the present study showed a statistically significant increase in the effective dose rate and lens dose rate, which were calculated by correcting the effective dose and lens equivalent dose with the fluoroscopy time (effective dose rate: *p* = 0.033; lens equivalent dose rate: *p* = 0.003) (Table 5).

The following two factors are the main reasons why the effective dose rate and lens equivalent dose rate increased:Improved use rate of personal passive dosimetersThe personal passive dosimeter management method shifted from single-dosimeter to double-dosimeter.

An increased use rate in personal passive dosimeters will result in the addition of exposure doses that were not considered in the values up to that point, thereby increasing the effective dose rate and lens equivalent dose rate. Furthermore, the personal passive dosimeter was managed with the single-dosimeter approach, where E = D_main_ according to Equation (2) and H_eye_ = D_main_ according to Equation (4). However, when the personal passive dosimeters were managed with the double-dosimeter approach and assuming the shielding rate by the lead apron to be 90%, E = 0.89 × D_main_ + 0.11 × D_additional_ = 1.99 × D_main_ from Equation (1) and H_eye_ = D_additional_ = 10 × D_main_ from Equation (3), thereby potentially increasing the effective dose rate and lens equivalent dose rate values. Chida et al. reported that the mean effective dose and lens equivalent dose were increased by 2.9- and 19.8-times, respectively, as the dosimeter management method was changed from a single-dosimeter to a double-dosimeter approach [36]. Similarly, for physicians g, h, i, and j in the present study (Appendix A), switching from a single-dosimeter to a double-dosimeter approach during the intervention period resulted in an increased mean effective dose rate and lens equivalent dose rate by 4.4- and 18.9-times, respectively (mean effective dose rate: 16.9/3.8 = 4.4, mean lens equivalent dose rate: 83.0/4.4 = 18.9).

The visual inspection survey of the use rate that was conducted in the present study was not 100% accurate; there is a bias in that the inspector was present during the daytime but not during the nighttime during an emergency. The sampling survey was conducted in a single facility. Hence, a limitation of the present study is that the use rate shown here is not necessarily reflective of the current situation in Japan. Different results may be obtained depending on other countries’ medical cultures and trends. Furthermore, the possibility of increasing awareness toward radiation protection among the physicians cannot be ruled out because the intervention aimed to reduce the effective dose rate and lens equivalent dose rate by lowering the fluoroscopy time, tightening the irradiation field, and actively using ceiling-suspended radiation shielding screens; however, these actions cannot be excluded from the obtained data. Furthermore, it is still unknown how stopping interventions by briefings after the end of this study would change the effective dose rate and lens equivalent dose rate. Regardless of whether the use of personal passive dosimeters or PPE could be rooted as part of a “protection culture,” careful follow-up observations are needed in the future. Implementing technology that automatically inspects and confirms the proper use of dosimeters and PPE might be indispensable to extend the results of this study.

## 5. Conclusions

A visual inspection of the use rate of personal passive dosimeters and PPE by IR physicians in this study showed that the use rate of this equipment was low overall, except for lead aprons. However, briefings were conducted immediately before the start of IR, and verbal confirmations were obtained on whether physicians correctly wore personal passive dosimeters and PPE as interventions; this dramatically increased the use rate of this equipment to close to 100%. However, the effective dose rate and lens equivalent dose rate corrected with the fluoroscopy time increased immediately after the intervention. This result was hypothesized to be due to the inclusion of exposure that has been overlooked to date. Additionally, the changes in the dose calculation method accompanied the difference in the personal passive dosimeter management method from a single-dosimeter approach to a double-dosimeter approach, which also had an effect on the effective dose rate and lens equivalent dose rate.

## Figures and Tables

**Figure 1 ijerph-19-16825-f001:**
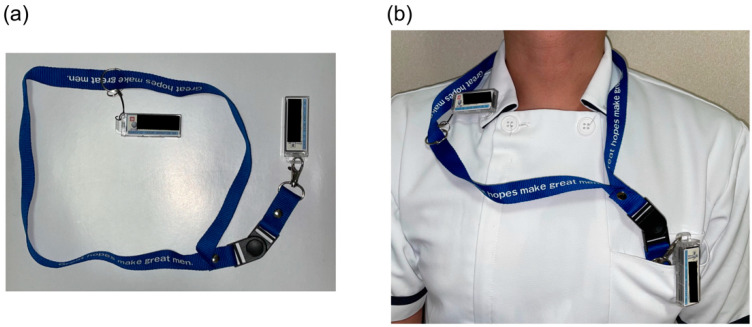
A unique initiative of the surveyed hospital. (**a**) A strap connects both the main and additional dosimeters. (**b**) The use of this strap prevents physicians from forgetting to wear the additional dosimeter.

**Table 1 ijerph-19-16825-t001:** Occupational health management in radiation medical care.

Forms of Occupational Health Management	Management Content
General management	Construction of safety management system related to radiation medical care, etc.
Work environment management	Measurement of radiation doses in the radiation clinic, personal monitoring implementation, etc.
Work management	Preparation of radiation treatment procedure manuals, maintenance of PPE *, etc.
Health management	Implementation of special radiation health examinations, etc.
Occupational health training	Regular education and training before and after placement of radiation work, etc.

* PPE: personal protective equipment.

**Table 2 ijerph-19-16825-t002:** Past reports on the use rate of PPE by physicians.

Authors	Country	Physicians’ Personal Protective Equipment Use Rate (%)			
		Personal Passive Dosimeters	Lead Aprons	Lead Glasses	Thyroid Protection Collars
Niklason et al. (1993) [10]	N/A *	40	N/A *	10	47
Tsapaki et al. (2009) [11]	Algeria, Kenya, Morocco, Sudan, Tunisia, Kuwait, Lebanon, Syria, Thailand, United Arab Emirates, Pakistan, Armenia, Bosnia and Herzegovina, Bulgaria, Croatia, Greece, Lithuania, Moldova, Slovenia, Tajikistan	96	100	73	N/A*
Efstathopoulos et al. (2011) [16]	Greece	N/A *	100	71–83	100
Vanhavere F et al. (2012) [17]	Belgium, Greece, France, Switzerland, Poland, Slovakia	N/A *	98–100	31–36	91–92
Vano et al. (2013) [12]	Argentine Republic	48–52	N/A *	41–52	N/A *
Lynskey III et al. (2013) [18]	N/A *	N/A *	99.4	54.2	94
International Atomic Energy Agency (2014) [13]	Global	70–77	97	24–47	N/A *
Brun et al. (2018) [14]	France	45.7–54.0	88–97.1	0–4	40–62.9
Kunugita (2019) [7]	Japan	17–100	N/A *	N/A *	N/A *
Altintas et al. (2020) [15]	N/A *	17.3	96.2	32.7	80.8

* N/A: not available.

**Table 3 ijerph-19-16825-t003:** Research Design.

Survey Content		Pre-Intervention Period (April 2017–August 2017)	Intervention Period (October 2017–February 2018)
All radiological treatments conducted by survey target physicians		549 cases	415 cases
Visual inspection of use status		Conducted by an inspector during the daytime	Conducted by an inspector at the pre-operative briefing during the daytime The inspector obtained verbal conformation and encouraged the physicians to wear proper equipment
Personal passive dosimeter	Main dosimeter	340 cases (61.9%)	321 cases (77.3%)
	Additional dosimeter		
Lead apron			
Lead glasses			
Thyroid protection collar			
Medical record survey		Conducted by researchers	Conducted by researchers
Personal exposure dose	Effective dose	549 cases (100%)	415 cases (100%)
	Lens equivalent dose		
Exposure-related indicators	Number of times radiation medical treatment was conducted		
	Fluoroscopy time		

**Table 4 ijerph-19-16825-t004:** Use rate of personal passive dosimeters and * PPE by physicians (*n* = 14).

Personal Passive Dosimeter/PPE	Pre-Intervention Period		Intervention Period		*p* Value ^†^
	Mean (%)	Median [Range] (%)	Mean (%)	Median [Range] (%)	
Main dosimeter (under the lead apron)	47	57 [0–100]	97	100 [67–100]	0.002
Additional dosimeter (over the lead apron)	30	0 [0–100]	65	85 [0–100]	0.008
Lead apron	100	100	100	100	1.000
Lead glasses	37	15 [0–100]	97	100 [67–100]	0.003
Thyroid protection collar	52	69 [0–100]	97	100 [89–100]	0.008

* PPE: personal protective equipment; ^†^ Wilcoxon’s signed rank test.

**Table 5 ijerph-19-16825-t005:** Personal exposure dose and dose rate for physicians (*n* = 14).

Personal Exposure Dose/Dose Rate	Pre-Intervention Period		Intervention Period		*p* Value ^†^
	Mean	Median [Range]	Mean	Median [Range]	
Effective dose (mSv)	1.1	0.4 [0.0–6.3]	1.4	1.1 [0.0–4.8]	0.345
Effective dose rate * (μSv/min)	2.3	1.3 [0.0–13.6]	7.3	3.1 [0.0–33.0]	0.033
Lens equivalent dose (mSv)	3.2	0.7 [0.0–23.7]	4.2	1.7 [0.1–13.9]	0.124
Lens equivalent dose rate ** (μSv/min)	3.7	2.4 [0.0–15.9]	28.7	7.5 [3.1–167.0]	0.003

* The effective dose rate for each physician was calculated as the Effective dose (mSv)/Total fluoroscopy time (min) in Appendix A. ** The lens equivalent dose rate for each physician was calculated as the Lens equivalent dose (mSv)/Total fluoroscopy time (min) in Appendix A. ^†^ Wilcoxon’s signed rank test.

## Data Availability

The data are not publicly available due to ethical issues.

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
