# Peer review of "The Effect of Pre-Operative Verbal Confirmation for Interventional Radiology Physicians on Their Use of Personal Dosimeters and Personal Protective Equipment"

_ijerph, 2022, doi:10.3390/ijerph192416825_

Round 1
Reviewer 1 Report (Previous Reviewer 4)
Dear Authors,
I see that you made some important improvements in your manuscript. As I wrote in the previous round, the topic is very interesting and the study, althoug with a small sample, well conducted. I just suggest to add a more simple description of the intervention (also by a scheme).
Best regards
Author Response
Please see the attachment.

Reviewer 2 Report (New Reviewer)
Summary
The paper ‘The effect of pre-operative verbal confirmation for interventional radiology physicians on their use of personal dosimeter and personal protective equipment’ by Matsuzaki et al. is well written and organized. Methodology seems to be adequate to the aim of the research with regard to use rates of PPE and dosimeters, but not with regard to exposure (effective dose (rate) and lens equivalent dose (rate)). Results are provided in the form of understandable tables, but the choice of summary measures could be improved. The work aims to provide information on the effect of visual inspection of personal protective equipment (PPE) and dosimeters on use rates and exposure dose of physicians working in interventional radiology (IR). The research could be helpful for increasing occupational safety for physicians working in IR.
General comments:
The authors present values for effective dose rate and lens equivalent dose rate. However, the changes between pre-intervention and intervention period in these measured values primarily reflect the change in the use of the measuring equipment (dosimeters) and not an actual change in the underlying quantities to be measured. Therefore, I see little point in presenting these measured values. They are essentially a proxy for the use rate, which is already presented separately. There are also some inconsistencies in the numbers. For example, for physicians h and i, Table S3 indicates that their effective dose rate and lens equivalent dose rate were 0 before the intervention, even though these physicians performed examinations. Thus, this can only be because the dosimeters were not worn (which is particularly odd, since Table S1 indicates that they both wore their main dosimeter at least once during this period), but this definitely does not mean that there was no exposure. I would recommend limiting the presentation of results to use rates.
Specific comments:
Line numbers refer to the clean version (track changes mode not activated).
Line 47-49: Please state explicitly the dose limits for general and ocular exposure and provide references for those limits.
Line 52, 55: LNT: The authors don’t need to introduce this abbreviation, they don’t use it again in the paper. ALARA: This abbreviation is also not used further. I guess the authors wanted to state it because it is a common abbreviation that many readers know. Perhaps the authors could write something like "… as low as reasonably achievable (commonly known as ALARA)."
Table 1: According to line 60, the first entry in column one should be “general management”. The abbreviation PPE is not explained in the table.
Table 2: I don’t think the column “journal” is needed here. One can easily get this information from the references. In my opinion, it would be more suitable to include information on the country the studies were conducted in.
Line 147: please write three hundred twenty-one as 321.
Table 3, line 141-152, line 373-74: The authors do not provide an explanation for the fact that the visual inspection of the use status was not performed in 100% of the cases. I assume that this was because there was only one investigator for this and accordingly this person was not always available. I wonder if there is a bias introduced here in that the investigator was (presumably) more likely to be present during the day and possibly at night, when emergencies are more likely to be treated, the use rate of dosimeters and personal protective equipment is different. I think it would be good to address this point in the discussion as well.
Line 189, equation (1): Please explain your choice of the weighting factors.
Equations (1)-(4): As described above, I think by using different formulas for calculating effective dose and lens equivalent dose depending on how many dosimeters were used, the values are not comparable anymore.
Table 4 & Table 5: I think it’s not really meaningful to give mean ± SD for these very skewed distributions. For example, the authors could present lower and upper quartile to give a better impression of the distributions if they feel presenting median and range is not sufficient.
Tables S1-S3: I’m quite concerned about data privacy regarding these tables. I think it is well possible to link these “anonymized” data to specific people. This is especially true for the one female physician included in the tables, because the authors surveyed all physicians conducting IR in this hospital at the time (see line 137).
Author Response
Please see the attachment.

Reviewer 3 Report (New Reviewer)
Reviewer comments
Title: The effect of pre-operative verbal confirmation for interventional radiology physicians on their use of personal dosimeter and personal protective equipment
Overall comments
It was difficult to review this paper as it was submitted with tracked changes and I had not seen the initial submission. In the future, it will be useful to ensure that the submitted article does not have tracked changes or that both versions (tracked changes and accepted changes) are made available.
This study was conducted from April 2017 – February 2018 but is only being submitted for publication in 2022. What led to such a delay and the big question is whether the data presented is relevant after about years have elapsed after the data collection. Further, the ethics approval was only granted in December 2017, how did the study proceed in April 2017 before the ethics approval was received?
The data availability statement indicates that this study did not create any new data, yet the study is reporting on data that was collected.
This paper will benefit from significant revision of the abstract, introduction, and materials & methods. Grammar can also be improved.
Below are the comments specific to the sections of the manuscript.
Title
It is difficult to comment in full whether the title of this paper is adequate because the aim of the paper has been written poorly.
Abstract
The abstract is unclear regarding the methodology used to conduct this research. The bulk of the abstract presents the results with little information about the methodology and just one sentence about the conclusion. The latter could be expanded.
Introduction
Overall, the introduction could be concise with some of the information left out and also have a clearly stated aim of the research. Currently, the aim of this study is poorly written.
Line 48 uses the word “radioscopy”, even though this word is correct, it is not a commonly used term. The use of this word in that sentence also creates tautology because it is a common understanding that interventional radiology uses x-rays. The meaning of the sentence will remain accurate if the words “using radioscopy” are taken out.
The sentence on lines 62 -63 about the risks, is written very poorly with no meaning.
There are quite a few statements made in the introduction without any supporting literature sources. For example, the statement about the requirement to attend radiation protection lectures for one hour/ per year, where did this come from?
Table 1 needs to be referenced.
Line 130 says “the investigator” yet the paper is written by a group of authors. The same applies to lines 176 and 180 where the description says “we surveyed” and then refers to an investigator.
Line 130 refers to “accurate fact-finding” – this terminology is not generally used for scientific writing.
Materials and Methods
Some of the information contained in this section belongs to the result section, e.g. the participants’ demographic data like age, gender, etc.
It is unclear who conducted the visual inspection for the use of PPE and dosimeters. It is also unclear how was this done with influencing the normal practice of the physicians because it is generally acknowledged that when individuals know they are being observed their practice may change especially in areas like radiation protection.
Table 3 does not add any value and can simply be deleted.
2.2 Radiation equipment – also does not add value.
Results and discussion
The results and discussion are probably written sufficiently but still need some of the demographic information that is in the materials and methods to be moved to the results section.
Author Response
Please see the attachment.

This manuscript is a resubmission of an earlier submission. The following is a list of the peer review reports and author responses from that submission.
Round 1
Reviewer 1 Report
The purpose of this study is to analyze the verbal confirmation effect based on a survey on the frequency of wearing a passive dosimeter and PPE in preintervention and intervention for interventional radiology physicians.
- Some contents of Tables 4, 5, and 6 are not presented and need to be corrected. Also, it seems necessary to minimize the table.
- The number of data samples is too small to be statistically significant. How about increasing the volume of your survey?
- Conclusions may be limited to certain countries and to some hospital interventional radiology physicians. Results may vary depending on medical culture and trends with other countries.
- As shown in Figure 3, it seems that more measures are needed to increase the wear of dosimeter and PPE.
- In the table 6, lens equivalent dose rate of i, j is too much and please check.
Reviewer 2 Report
The topic of the paper, and its research questions, are highly relevant and of great practical interest.
The paper demonstrates knowledge of comparable methods presented in the literature.
The case study is an important part of the paper. The proposed methodology is very interestig.
Some issues that may enhance the contribution of this work:
- the authors should better explain the figure 1;
- the authors should better explain the questionnarie structure and the reasons (it is a highly relevant part of your paper)
- the authors should correct some grammatical errors
Please improve english and references, e.g.:
- Systematic Human Reliability Analysis (SHRA): A New Approach to Evaluate Human Error Probability (HEP) in a Nuclear Plant Di Bona, G.D., Falcone, D., Forcina, A., Silvestri, L. International Journal of Mathematical, Engineering and Management Sciences, 2021, 6(1), pp. 345–362
- Quality Checks Logit Human Reliability (LHR): A New Model to Evaluate Human Error Probability (HEP) Di Bona, G., Falcone, D., Forcina, A., De Carlo, F., Silvestri, L. Mathematical Problems in Engineeringthis link is disabled, 2021, 2021, 6653811
- Dependence assessment in human reliability analysis under uncertain and dynamic situations X Gao, X Su, H Qian, X Pan - Nuclear Engineering and Technology, 2022 - Elsevier
- Robust data-driven human reliability analysis using credal networks C Morais, HD Estrada-Lugo, S Tolo, T Jacques Reliability Engineering & System Safety Volume 218, Part A, February 2022, 107990
Reviewer 3 Report
The article describes an action to raise awareness among Interventional radiology physicians in order to increase the use of dosimeters and PPE.
Despite being a subject of interest to clinical practice, the way the intervention is presented does not seem to have enough innovation to be relevant to the scientific. It is a common intervention in several areas at the OSH level, presenting the expected results.
Some comments:
-Title is too long. An ideal title should not exceed 100 characters;
- The objective is not clearly defined;
- Recurring use of the personal pronoun "we". The use of personal pronouns should be avoided in scientific language;
-The sample is too small;
-Tables 3, 4 and 5 are incomplete in the manuscript;
- The methodology lacks a more detailed explanation;
-Figures 1 and 2 are not necessary. Their information could be synthesized;
- The results show high standard deviations that are not explained;
Reviewer 4 Report
Dear Authors,
I have been very pleased to read your work. You targeted a main occupational health issue. Please take note of my comments which I hope may help you improving your manuscript.
Introduction: it would be interesting to provide a framework of diseases caused by work-realted radiation exposure in Japan and why there is the need to use a rigid PPE protocol regarding this occupational hazard (I think it is worth mentioning the fact that there is no dose value below it is possible to rule out a health risk)
Sample: you should justify your sample in some way, maybe referring to the volume of interventions of the hospital and to the number of interventional radiologists totally employed.
line 123: could you please indicate also gender distribution?
line 135-136: it is not clear who perform the visula ispection and overall the part of analysis of the work condition and use of PPE.
You have to be clearer about the intervention, you could add a small paragraph in the methods section
figure 2: it is unreadable, consider to put in supplementary material
table 4,5 and 6: I suppose for difficulty in layout but some parts are missing, please check and consider to split the table in two.
Discussion: could you think about an effective solution/intervention to sustain the results obtained?
Best wishes and good luck with your work